# Derivation and validation of the J-CTO extension score for pre-procedural prediction of major adverse cardiac and cerebrovascular events in patients with chronic total occlusions

**Soichiro Ebisawa**[1]*, **Shun Kohsaka**[2], **Toshiya Muramatsu**[3], **Yoshifumi Kashima**[4], **Atsunori Okamura**[5], **Masahisa Yamane**[6], **Masami Sakurada**[7], **Shunsuke Matsuno**[8], **Mikihiro Kijima**[9], **Maoto Habara**[10]

1 Department of Cardiovascular Medicine, Shinshu University School of Medicine, Matsumoto, Japan, 2 Department of Cardiology, Keio University School of Medicine, Tokyo, Japan, 3 Cardiovascular Center, Tokyo General Hospital, Tokyo, Japan, 4 Division of Cardiology, Sapporo Cardio Vascular Clinic, Hokkaido, Japan, 5 Division of Cardiology, Sakurabashi-Watanabe Hospital, Osaka, Japan, 6 Cardiology Department, Saitama Sekishinkai Hospital, Saitama, Japan, 7 Department of Cardiology, Tokorozawa Heart Center, Saitama, Japan, 8 Department of Cardiovascular Medicine, The Cardiovascular Institute, Tokyo, Japan, 9 Cardiology and Vascular Medicine, Hoshi General Hospital, Fukushima, Japan, 10 Department of Cardiovascular Medicine, Toyohashi Heart Center, Aichi, Japan

* ebifender@yahoo.co.jp

## Abstract

We developed a prediction model of long-term risk after percutaneous coronary intervention (PCI) for coronary chronic total occlusion (CTO) based on pre-procedural clinical information. A total of 4,139 eligible patients, who underwent CTO-PCI at 52 Japanese centers were included. Specifically, 1,909 patients with 1-year data were randomly divided into the derivation (n = 1,273) and validation (n = 636) groups. Major adverse cardiac and cardiovascular event (MACCE) was the primary endpoint, including death, stroke, revascularization, and non-fatal myocardial infarction. We assessed the performance of our model using the area under the receiver operating characteristic curve (AUC) and assigned a simplified point-scoring system. One-hundred-thirty-eight (10.8%) patients experienced MACCE in the derivation cohort with hemodialysis (HD: odds ratio [OR] = 2.55), left ventricular ejection fractions (LVEF) <35% (OR = 2.23), in-stent occlusions (ISO: OR = 2.27), and diabetes mellitus (DM: OR = 1.72). The AUC of the derivation model was 0.650. The model's performance was similar in the validation cohort (AUC, 0.610). When assigned a point for each associated factor (HD = 3, LVEF <35%, ISO = 2, and DM = 1 point), the average predicted versus the observed MACCE probability using the Japan-CTO extension score for the low, moderate, high, and very high risk groups was 8.1% vs. 7.3%, 16.9% vs. 15.9%, 22.0% vs. 26.1%, and 56.2% vs. 44.4%, respectively. This novel risk model may allow for the estimation of long-term risk and be useful in disseminating appropriate revascularization procedures.

**Data Availability Statement:** All relevant data are within the manuscript and its Supporting Information files.

**Funding:** Soichiro Ebisawa belongs to Endowed Department of Cardiovascular Medicine of Shinshu University supported by Medtronic Japan Co.,Ltd. Abbott Vascular Japan Co.,Ltd. Boston Scientific Japan, TERUMO CORPORATION, Cardinal Health Japan and NIPRO CORPORATION. The funders provided support in the form of salaries for Soichiro Ebisawa, but did not have any additional role in the study design, data collection and analysis, decision to publish or preparation of the manuscript.

**Competing interests:** Soichiro Ebisawa belongs to Endowed Department of Cardiovascular Medicine of Shinshu University supported by Medtronic Japan Co.,Ltd. Abbott Vascular Japan Co.,Ltd. Boston Scientific Japan, TERUMO CORPORATION, Cardinal Health Japan and NIPRO CORPORATION. This does not alter our adherence to PLOS ONE policies on sharing data and materials.

## Introduction

While percutaneous coronary intervention (PCI) remains a valid treatment option for patients with chronic total occlusion (CTO), the outcomes of patients following PCI vary significantly among medical centers. Additionally, the optimal treatment strategies remain controversial despite the technological and methodological advances [1,2]. Particularly, the high incidence of major adverse cardiac and cerebrovascular events (MACCEs) remains problematic. Under the current treatment protocols, over 20.7% of patients who undergo CTO-PCI experience restenosis within 5 years, which is considerably higher compared to the rate of those who have not undergone this procedure [3]. Conversely, the long-term outcome for coronary artery bypass grafting (CABG) for CTO remains favorable, with over 90% patency of the left internal mammary artery [4]. Standardized pre-procedural risk assessment tools may be helpful in identifying patients at high-risk for developing MACCE after CTO-PCI. In particular, identification of patients at risk for restenosis at the treatment selection time would aid in pre-procedural decision-making.

Historically, as CTO-PCI procedures are complex [5–7], requiring high contrast media volume and radiation during the procedure [8,9], risk models for procedural success have been developed [10–12]. However, these risk models do not effectively predict long-term outcomes following the procedure. Recently, the Prospective Global Registry for the Study of CTO Intervention (PROGRESS CTO) [11] score, developed to predict the success of CTO-PCIs, was reported to be a useful tool in predicting long-term outcomes [13], albeit the study population was small and the procedural success rates were lower than that in conventional CTO studies [7].

Therefore, our aim was to develop a pre-procedural risk score for patients with CTO using a multi-institutional dataset from Japan (the Retrograde Summit Registry). Specifically, utilizing pre-procedural information on patients' backgrounds and angiograms, we compared the performance of the novel outcome-specific risk scores with the traditional procedural scores (Japan-CTO [J-CTO]).

## Materials and methods

This study was approved by the ethic committee of all participating facilities of retrograde summit registry (Toyohashi Heart Center, Saitama Sekishinkai Hospital, Tokyo General Hospital, Sakurabashi-Watanabe Hospital, Sapporo Cardio Vascular Clinic, The Cardiovascular Institute, Tokorozawa Heart Center, Hoshi General Hospital, Kyoto Okamoto Memorial Hospital, Saitama Prefecture Cardiovascular and Respiratory Center, Mie Heart Center, Takase Clinic, Nagoya Heart Center, Higashi Takarazuka Satoh Hospital, Hokkaido Social Insurance Hospital, Shiga Medical Center for Adults, Kakogawa East City Hospital, Hokko Memorial Hospital, Yotsuba Circulation Clinic, Yokohama Sakae Kyosai Hospital, Saiseikai Fukuoka General Hospital, Edogawa Hospital, Fukaya Red Cross Hospital, Rinku General Medical Center, Nagoya Daini Red Cross Hospital, Hyogo Prefectural Amagasaki Hospital, Sanda City Hospital, Itabashi Chuo Hospital, Nagoya Tokushukai Hospital, Showa General Hospital, Kanagawa Cardiovascular and Respiratory Center, Seirei Hamamatsu General Hospital, Tokeidai Memorial Hospital, Kyoto Katsura Hospital, Kushiro City General Hospital, Iwate Prefectural Central Hospital, Kusatsu Heart Center, Hamada Medical Center, Tokuyama Chuo Hospital, Showa University Hospital, Osaka Saiseikai Izuo Hospital, Todachuo General Hospital, Nozaki Tokushukai Medical Center, Shinkoga Hospital, Mito Brain Heart Center, Shuuwa General Hospital, Iwaki Kyouritsu Hospital, Hyogo Brain and Heart Center, NTT East Sapporo Hospital, Chikamori Hospital, Mimihara General Hospital, Hokusetsu General Hospital, Kobe Redcross Hospital, Kansai Medical University Takii Hospital, Tokushima Red

Cross Hospital, Osaki Citizen Hospital, Tsukuba Memorial Hospital, Yokohama Shintoshi Neurosurgical Hospital, Fukuoka City Hospital, Bellland General Hospital, Matsubara Tokushukai Medical Center, Ohta General Hospital Ohta Nishinouchi Hospital, Sapporo Orthopaedics and Cardiovascular Hospital, Toho University Omori Medical Center) and review board in Shinshu University. Written informed consent was obtained from all participants. All procedures performed in studies involving human participants were in accordance with the 1964 Helsinki declaration and its later amendments or comparable ethical standards.

The Retrograde Summit was constructed as a multicenter, prospective, nonrandomized registry of patients treated at 65 Japanese centers between January 2012 and December 2015. Within the study period, 4,909 patients with CTO lesions underwent elective PCIs and had useful initial data. We excluded 770 patients who underwent CTO-PCI in 13 institutions that did not join to collect the 1-year follow-up data, and the final number of baseline participants for the analysis was 4,139. Among them, 1,909 patients had 1-year follow-up data (Fig 1). Emergent cases were excluded from this study. The indication of CTO-PCI or bypass grafting was determined via discussions among the cardiac team at each institution. The PCI-related strategies (e.g., retrograde or antegrade approach, stent deployment, or nothing) depended upon the operators' discretion.

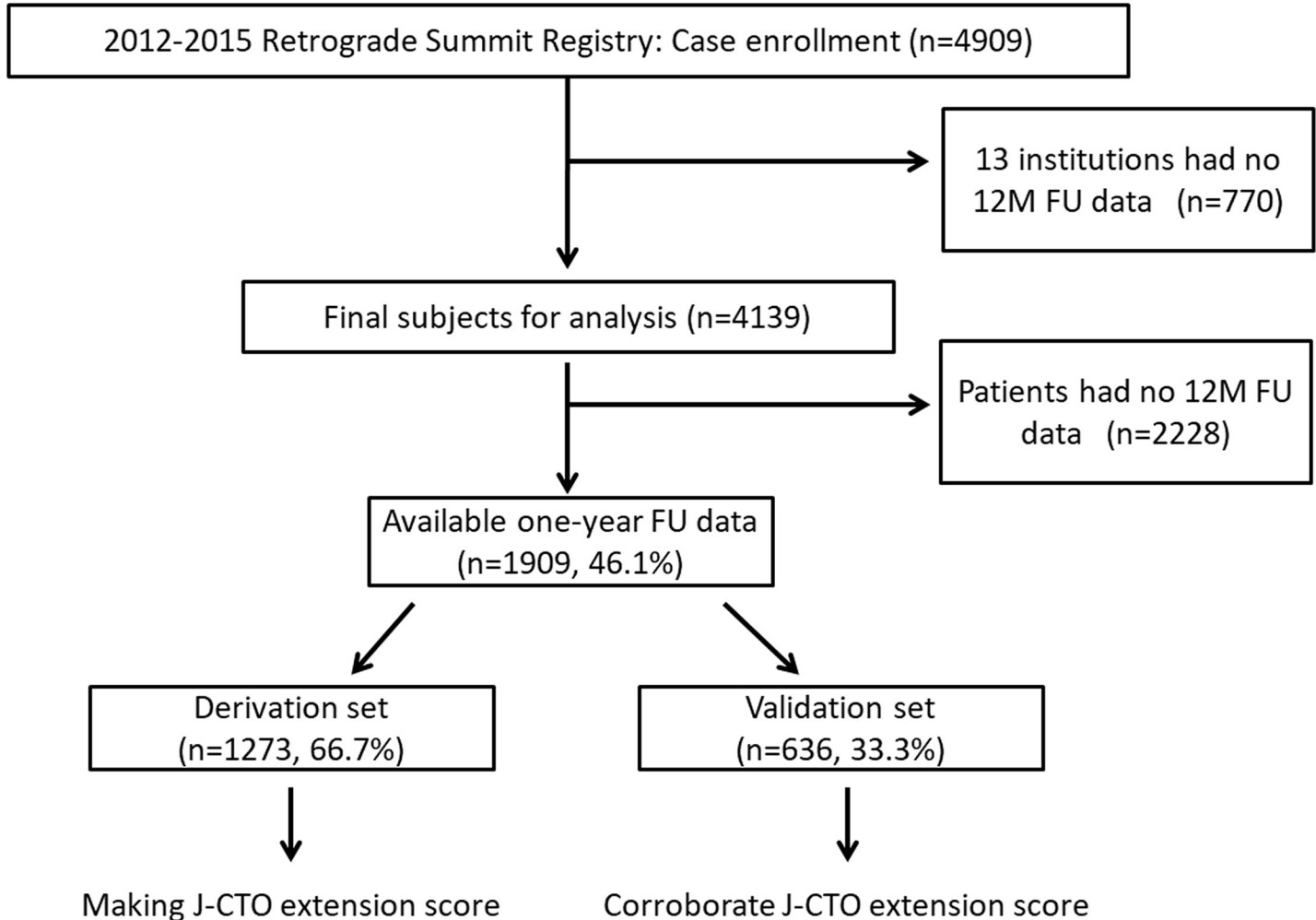

**Fig 1. Study flow.** When the patients had CTO lesions in several vessels, only the vessel that was treated first was analyzed.

The baseline patient characteristics, procedural details and techniques, and in-hospital outcomes were obtained. Procedural success was defined as the guidewire and balloon crossing a completely occluded lesion resulting in the successful dilation of the occluded artery, and restoration of antegrade flow (thrombolysis in myocardial infarction flow grade 3) with <50% residual stenosis on final angiography. The primary endpoint of this analysis was a composite of major adverse cardiac and cerebrovascular events (MACCE) observed at the 1-year follow-up. One-year MACCE was defined as death, myocardial infarction, stroke, and any target lesion revascularization [14].

Neither a centralized event adjudication nor core laboratory assessment was performed. All clinical events were reported by the operator who performed the CTO-PCI procedure.

CTO was defined as a complete occlusion with thrombolysis in myocardial infarction flow grade 0, antegrade through the affected segment, and >3 months in duration, by the opinion of the operator (based on clinical features, angiographic features, and/or previous imaging results). Chronic kidney disease was defined as an estimated glomerular filtration rate <60 ml/min/1.73 m$^2$, calculated using the diet modification in renal disease formula.

The angiographic morphology of the entry point was classified as "blunt" when the occluded segment did not end in a funnel tapered form. Lesion calcification was assigned to one of three categories: mild (spots), moderate, and severe (involving ≤50% and >50% of the reference lesion diameter, respectively). Lesion bending was defined as at least one bend of >45˚, assessed by angiography, throughout the occluded segment. Proximal vessel tortuosity was defined as the presence of one bend of >70˚ in the CTO vessel. The occlusion length was categorized as <20 or ≥20 mm. The collateral connection grade was classified as previously described [15]. The lesion difficulty was classified using the J-CTO score. The retrograde approach was defined as any PCI attempt for CTO, with wiring through the collateral arteries, that did not depend on a successful pass through the collateral root.

We separately performed analyses in the population subsets that were used for derivation (2/3 random sampling group, 1,273 patients) and validation (1/3 random sampling rate, 636 patients) studies (Fig 2). A two-step analysis was used to identify the independent predictors of the 1-year MACCE. First, from the derivation cohort, univariate analysis was performed to identify the clinical and angiographical variables associated with the incidence of the 1-year MACCE. Second, we performed a multivariate logistic regression analysis to determine the configured variables of the final scoring model, the variables that were strongly (p <0.05) associated with the incidence of the 1-year MACCE, from the univariate analysis. In this model, low left ventricular function cases were treated as a categorical variable (left ventricular ejection fraction [LVEF] <35%) to increase the model's clinical utility. We scored each variable and calculated the overall score for each case, according to the odds ratio (OR) calculated by the multivariate analysis. Subsequently, the derivation and validation groups were divided to four subsets, defined as low, moderate, high, and very high groups according to the scoring model. These four subsets were used to confirm the differences of the stepwise alternations in the 1-year MACCE incidences between the derivation and validation groups.

The continuous and categorical variables are presented as means ± standard and as numbers and percentages, respectively. They were evaluated using two-sided, unpaired t-tests and using the chi-squared or Fisher's exact test, where appropriate, respectively. All reported p values were two-sided. The statistically significant level was set at p <0.05. A logistic regression model was performed to determine the predictors of the incidences of MACCE at 1 year after PCI. In the first step, several potential predictors were separately assessed by univariate logistic regression analyses. A multiple logistic regression analysis was, then, conducted using covariates associated with the 2-year incidence of MACCE in the univariate analyses (p ≤0.05). The results are expressed as OR with 95% confidence intervals (CI). Using the validation cohort

**TREND Statement Checklist**

| Paper Section/ Topic | Item No | Descriptor | Reported? ✓ | Pg # |
|---|---|---|---|---|
| **Title and Abstract** | | | | |
| Title and Abstract | 1 | • Information on how unit were allocated to interventions | ✓ | 3 |
| | | • Structured abstract recommended | ✓ | 3 |
| | | • Information on target population or study sample | ✓ | 3 |
| **Introduction** | | | | |
| Background | 2 | • Scientific background and explanation of rationale | ✓ | 5 |
| | | • Theories used in designing behavioral interventions | ✓ | 5 |
| **Methods** | | | | |
| Participants | 3 | • Eligibility criteria for participants, including criteria at different levels in recruitment/sampling plan (e.g., cities, clinics, subjects) | ✓ | 6 |
| | | • Method of recruitment (e.g., referral, self-selection), including the sampling method if a systematic sampling plan was implemented | ✓ | 6 |
| | | • Recruitment setting | ✓ | 6 |
| | | • Settings and locations where the data were collected | ✓ | 6 |
| Interventions | 4 | • Details of the interventions intended for each study condition and how and when they were actually administered, specifically including: | ✓ | 7 |
| | | ○ Content: what was given? | ✓ | 7 |
| | | ○ Delivery method: how was the content given? | ✓ | 7 |
| | | ○ Unit of delivery: how were the subjects grouped during delivery? | ✓ | 7 |
| | | ○ Deliverer: who delivered the intervention? | ✓ | 7 |
| | | ○ Setting: where was the intervention delivered? | ✓ | 7 |
| | | ○ Exposure quantity and duration: how many sessions or episodes or events were intended to be delivered? How long were they intended to last? | · | |
| | | ○ Time span: how long was it intended to take to deliver the intervention to each unit? | · | |
| | | ○ Activities to increase compliance or adherence (e.g., incentives) | | |
| Objectives | 5 | • Specific objectives and hypotheses | | |
| Outcomes | 6 | • Clearly defined primary and secondary outcome measures | ✓ | 6 |
| | | • Methods used to collect data and any methods used to enhance the quality of measurements | ✓ | 6 – 7 |
| | | • Information on validated instruments such as psychometric and biometric properties | | |
| Sample Size | 7 | • How sample size was determined and, when applicable, explanation of any interim analyses and stopping rules | ✓ | 6 |
| Assignment Method | 8 | • Unit of assignment (the unit being assigned to study condition, e.g., individual, group, community) | ✓ | 6 |
| | | • Method used to assign units to study conditions, including details of any restriction (e.g., blocking, stratification, minimization) | | |
| | | • Inclusion of aspects employed to help minimize potential bias induced due to non-randomization (e.g., matching) | | |

**TREND Statement Checklist**

| Numbers analyzed | 16 | • Number of participants (denominator) included in each analysis for each study condition, particularly when the denominators change for different outcomes; statement of the results in absolute numbers when feasible | ✓ | 6 |
|---|---|---|---|---|
| | | • Indication of whether the analysis strategy was "intention to treat" or, if not, description of how non-compliers were treated in the analyses | | |
| Outcomes and estimation | 17 | • For each primary and secondary outcome, a summary of results for each estimation study condition, and the estimated effect size and a confidence interval to indicate the precision | ✓ | 10 – 11 |
| | | • Inclusion of null and negative findings | | |
| | | • Inclusion of results from testing pre-specified causal pathways through which the intervention was intended to operate, if any | | |
| Ancillary analyses | 18 | • Summary of other analyses performed, including subgroup or restricted analyses, indicating which are pre-specified or exploratory | ✓ | 12 |
| Adverse events | 19 | • Summary of all important adverse events or unintended effects in each study condition (including summary measures, effect size estimates, and confidence intervals) | ✓ | 11 |
| **DISCUSSION** | | | | |
| Interpretation | 20 | • Interpretation of the results, taking into account study hypotheses, sources of potential bias, imprecision of measures, multiplicative analyses, and other limitations or weaknesses of the study | ✓ | 12 |
| | | • Discussion of results taking into account the mechanism by which the intervention was intended to work (causal pathways) or alternative mechanisms or explanations | ✓ | 12 |
| | | • Discussion of the success of and barriers to implementing the intervention, fidelity of implementation | ✓ | 12 |
| | | • Discussion of research, programmatic, or policy implications | ✓ | 12 |
| Generalizability | 21 | • Generalizability (external validity) of the trial findings, taking into account the study population, the characteristics of the intervention, length of follow-up, incentives, compliance rates, specific sites/settings involved in the study, and other contextual issues | ✓ | 13 |
| Overall Evidence | 22 | • General interpretation of the results in the context of current evidence and current theory | ✓ | 13 |

From: Des Jarlais, D. C., Lyles, C., Crepaz, N., & the Trend Group (2004). Improving the reporting quality of nonrandomized evaluations of behavioral and public health interventions: The TREND statement. *American Journal of Public Health*, 94, 361-366. For more information, visit: http://www.cdc.gov/trendstatement/

**TREND Statement Checklist**

| Blinding (masking) | 9 | • Whether or not participants, those administering the interventions, and those assessing the outcomes were blinded to study condition assignment; if so, statement regarding how the blinding was accomplished and how it was assessed. | | |
|---|---|---|---|---|
| Unit of Analysis | 10 | • Description of the smallest unit that is being analyzed to assess intervention effects (e.g., individual, group, or community) | ✓ | 7 |
| | | • If the unit of analysis differs from the unit of assignment, the analytical method used to account for this (e.g., adjusting the standard error estimates by the design effect or using multilevel analysis) | | |
| Statistical Methods | 11 | • Statistical methods used to compare study groups for primary methods outcome(s), including complex methods of correlated data | ✓ | 8 |
| | | • Statistical methods used for additional analyses, such as a subgroup analyses and adjusted analysis | ✓ | 8 |
| | | • Methods for imputing missing data, if used | ✓ | 8 |
| | | • Statistical software or programs used | ✓ | 8 |
| **Results** | | | | |
| Participant flow | 12 | • Flow of participants through each stage of the study: enrollment, assignment, allocation, and intervention exposure, follow-up, analysis (a diagram is strongly recommended) | ✓ | 10 |
| | | ○ Enrollment: the numbers of participants screened for eligibility, found to be eligible or not eligible, declined to be enrolled, and enrolled in the study | ✓ | 10 |
| | | ○ Assignment: the numbers of participants assigned to a study condition | ✓ | 10 |
| | | ○ Allocation and intervention exposure: the number of participants assigned to each study condition and the number of participants who received each intervention | ✓ | 10 |
| | | ○ Follow-up: the number of participants who completed the follow-up or did not complete the follow-up (i.e., lost to follow-up), by study condition | ✓ | 10 |
| | | ○ Analysis: the number of participants included in or excluded from the main analysis, by study condition | ✓ | 10 |
| | | • Description of protocol deviations from study as planned, along with reasons | ✓ | 10 |
| Recruitment | 13 | • Dates defining the periods of recruitment and follow-up | ✓ | 6 |
| Baseline Data | 14 | • Baseline demographic and clinical characteristics of participants in each study condition | ✓ | 24 |
| | | • Baseline characteristics for each study condition relevant to specific disease prevention research | ✓ | 24 |
| | | • Baseline comparisons of those lost to follow-up and those retained, overall and by study condition | ✓ | 24 |
| | | • Comparison between study population at baseline and target population of interest | ✓ | 24 |
| Baseline equivalence | 15 | • Data on study group equivalence at baseline and statistical methods used to control for baseline differences | ✓ | 24 |

**Fig 2. TREND checklist.** We checked 22 points including TREND statement checklist. All points were included in our study.

data, the validities of the two risk models were also evaluated by examining the agreement between the predicted and observed proportions of the MACCE at 1 year after PCI, in the aforementioned four subsets, according to the total points. All analyses were performed using SPSS version 23.0 (IBM Corp., Armonk, NY, USA).

## Results and discussion

A total of 211 patients (11.1%) experienced MACCE at the 1-year follow-up, including 44 deaths, two myocardial infarctions, four strokes, and 162 target lesion revascularizations. Regarding the technical issues, a retrograde approach was performed in 606 cases (31.7%); however, this procedure was not associated with the initial success (p = 0.569) and the 1-year MACCE incidence rates (p = 0.368). The latter did not differ between the derivation and validation groups (10.8% vs 11.5%, p = 0.699). We compared the patients' baseline characteristics in the derivation and validation groups, and the results are presented in Table 1.

The mean age was 67.7±10.3 and 67.7±10.5 years in the derivation and validation groups (p = 0.367), respectively. All of the other patient's background variables did not differ between these groups. The angiographic characteristics did not differ between the two groups, and their calculated mean J-CTO scores were equivocal (1.53±1.09 vs 1.48±1.03, p = 0.443).

Table 2 shows the univariate analysis of the incidences of MACCE at the 1-year follow-up in the derivation group. Diabetes mellitus and LVEF <35% were significantly associated with the incidence of MACCE (60% vs 42.4%, p <0.0001 and 13.4% vs 6.3%, p = 0.006 between the patients with and without MACCE, respectively). The peripheral artery disease and hemodialysis histories were also significantly associated with MACCE (21.5% vs 11.3%, p = 0.0008; 20% vs 4.4%, p <0.0001, respectively). Regarding the lesion-related variables, in-stent occlusion lesions were associated with the incidence of MACCE (24.6% vs 13.3%, p <0.0001). Concerning the J-CTO score components, calcification and tortuous lesions were more often observed in the MACCE (+) group (40.4% vs 31.9%, p = 0.031; 13.8% vs 8.4%, p = 0.041). Conversely, the initial patient success was not associated with the incidence of the 1-year MACCE (p = 0.569).

According to the multivariate logistic regression analysis (Table 3) hemodialysis (OR, 2.552; 95% CI, 1.286–5.064; p = 0.007), LVEFs <35% (OR 2.233; 95% CI, 1.191–4.187; p = 0.012), in-stent occlusion lesions (OR, 2.279; 95% CI, 1.407–3.691; p = 0.001), and DM (OR, 1.722; 95% CI, 1.131–2.622; p = 0.011) were significant predictors of MACCE incidence at the 1-year follow-up. The derivation model area under the receiver operating characteristic (ROC) curve (AUC) was 0.650 (95% CI, 0.598–0.703; p <0.0001) and the Hosmer-Lemeshow test result was p = 0.632. The outcomes were similar in the validation cohort; the AUC was 0.610 (95% CI, 0.532–0.688; p = 0.003) and the Hosmer-Lemeshow test result was p = 0.720. We created a scoring model (J-CTO extension score) by assigning a weighted integer based on the calculated OR [16,17] (hemodialysis = 3, LVEF <35% = 2, in-stent occlusion = 2, and DM = 1). This model was successful in predicting the incidence of MACCE at the 1-year follow-up, with stepwise alterations in the derivation and validation groups (Fig 3). To assess the predictive power of this model, the proposed performance scoring system was compared with that of the traditional multivariate regression model (covariates selected from variables in Table 3) and only a small difference in C-statistics between them was observed (0.665 vs. 0.658).

The average predicted versus observed probabilities of MACCE with the J-CTO extension score for the low, moderate, high, and very high-risk subsets were as follows: 8.1% vs. 7.3%, 16.9% vs. 15.9%, 22.0% vs. 26.1%, and 56.2% vs. 44.4% of the observed and predicted scores, respectively (Fig 4A). The agreements between the observed and predicted risks of MACCE at

**Table 1. Comparison of the participants' baseline characteristics in the validation and derivation groups.**

| | Derivation Group | Validation Group | p value |
|---|---|---|---|
| **n** | 1,273 (%) | 636 (%) | |
| **Male sex** | 1,059 (83.5) | 529 (83.4) | 1 |
| **Age** | 67.7±10.3 | 67.7±10.5 | 0.367 |
| **Hypertension** | 985 (78.35) | 511 (80.5) | 0.142 |
| **Dyslipidemia** | 904 (71.8) | 468 (74.0) | 0.326 |
| **Diabetes mellitus** | 557 (44.3) | 284 (45.2) | 0.694 |
| **Smoking** | 641 (53.8) | 328 (53.7) | 0.098 |
| **Familial history** | 179 (19.9) | 88 (18.7) | 0.28 |
| **History of myocardial infarction** | 492 (40.1) | 262 (42.9) | 0.247 |
| **Post PCI** | 762 (60.1) | 386 (61.6) | 0.549 |
| **Post CABG** | 101 (8.0) | 57 (9.1) | 0.427 |
| **History of peripheral artery disease** | 148 (12.4) | 71 (11.8) | 0.601 |
| **Symptomatic** | 809 (65.9) | 409 (66.7) | 0.794 |
| **Hemodialysis** | 69 (5.6) | 36 (5.9) | 0.832 |
| **Chronic kidney disease** | 537 (42.6) | 272 (43.0) | 0.883 |
| **LVEF <35** | 86 (7.6) | 42 (6.9) | 1 |
| **Target lesion** | | | |
| **RCA** | 582 (45.9) | 307 (48.4) | 0.399 |
| **LAD LMT** | 406 (32.0) | 188 (29.6) | |
| **LCX** | 283 (22.3) | 138 (21.8) | |
| **In-stent occlusion** | 181 (14.5) | 91 (14.6) | 1 |
| **Reference size <2.5 mm** | 398 (33.3) | 197 (32.5) | 0.348 |
| **Collateral grade >2** | 310 (29.1) | 169 (31.2) | 0.483 |
| **Moderate to severe calcification** | 416 (32.8) | 215 (34.2) | 0.568 |
| **Retry case** | 135 (10.7) | 60 (9.5) | 0.471 |
| **Tortuous** | 114 (9.0) | 56 (8.8) | 0.932 |
| **Lesion length >20 mm** | 570 (53.2) | 272 (50.8) | 0.667 |
| **Blunt type** | 688 (54.4) | 333 (52.7) | 0.494 |
| **Value of J-CTO score** | 1.53±1.09 | 1.48±1.03 | 0.443 |

n, number of patients; PCI, percutaneous coronary intervention; CABG, coronary artery bypass graft; LVEF, left ventricular ejection fraction; RCA, right coronary artery; LMT, left main trunk; LAD, left anterior descending; LCX, left circumflex; CTO, chronic total occlusion

the 1-year follow-up, with developed risk-scoring methods were assessed across the 10 groups divided according to the risk score in the validation cohort. The correlation between the values of the observed and predicted risk in the 10 groups was significant (r = 0.77) (Fig 4B).

Fig 5A–5C shows the comparisons of the other predictive CTO scores in this population with the ROC curve. The J-CTO [10], Clinical and lesion-related (CL) [12], and PROGRESS CTO scores have all been previously developed to predict the initial success of CTO-PCI [11]. Fig 5A demonstrates a ROC curve of the whole patient population. The AUC of the J-CTO, CL, and PROGRESS CTO scores were 0.518 (95% CI, 0.473–0.563; p = 0.406), 0.540 (95% CI, 0.496–0.583; p = 0.066), and 0.514 (95% CI, 0.472–0.556; p = 0.509), respectively. The J-CTO extension score was only associated with the incidence of MACCE; the AUC was 0.634 (95% CI, 0.590–0.678; p <0.0001). Fig 5B and 5C shows ROC curves of the derivation and validation groups, respectively. In both groups, only the J-CTO extension score was associated with the incidence of MACCE at 1-year follow-up.

**Table 2. Univariate analysis for the incidences of MACCE in the derivation group.**

| | MACCE (+) | MACCE (-) | p value |
|---|---|---|---|
| **n** | 138 (%) | 1135 (%) | |
| **Men** | 119 (86.2) | 940 (83.1) | 0.329 |
| **Age ≥65 years** | 92 (66.6) | 743 (65.5) | 0.548 |
| **Hypertension** | 108 (80) | 877 (78.1) | 0.66 |
| **Dyslipidemia** | 96 (70.0) | 808 (72.0) | 0.616 |
| **Diabetes mellitus** | 81 (60) | 476 (42.4) | <0.0001* |
| **Smoking** | 74 (58.7) | 567 (53.2) | 0.276 |
| **Familial history** | 17 (17.7) | 162 (20.1) | 0.811 |
| **History of myocardial infarction** | 58 (43.6) | 434 (39.7) | 0.4 |
| **Post PCI** | 86 (62.3) | 676 (59.9) | 0.645 |
| **Post CABG** | 14 (10.2) | 87 (7.7) | 0.314 |
| **History of peripheral artery disease** | 28 (21.5) | 120 (11.3) | 0.008* |
| **Symptomatic (CCS ≥1)** | 88 (67.6) | 721 (65.7) | 0.696 |
| **Hemodialysis** | 21 (20) | 48 (4.4) | <0.0001* |
| **Chronic kidney disease** | 65 (49.6) | 472 (41.8) | 0.094 |
| **LVEF <35** | 18 (13.4) | 68 (6.3) | 0.006* |
| **Target lesion LCX** | 36 (26.0) | 247 (21.7) | 0.278 |
| **In-stent occlusion** | 33 (24.6) | 148 (13.3) | 0.001* |
| **Reference size <2.5 mm** | 46 (37.0) | 352 (32.9) | 0.118 |
| **Collateral grade >2** | 123 (95.3) | 1035 (96.8) | 0.991 |
| **Moderate to severe calcification** | 55 (40.4) | 361 (31.9) | 0.031* |
| **Retry** | 9 (6.5) | 126 (11.2) | 0.108 |
| **Tortuous** | 19 (13.8) | 95 (8.4) | 0.041* |
| **Lesion length >20mm** | 65 (59.6) | 505 (52.4) | 0.087 |
| **Blunt type** | 68 (49.2) | 620 (55.1) | 0.205 |
| **Initial procedural success** | 118 (85.5) | 1012 (89.1) | 0.200 |

n, number of patients; CCS, Canadian Cardiovascular Society grade; PCI, percutaneous coronary intervention; CABG, coronary artery bypass graft; LVEF, left ventricular ejection fraction; LCX, left circumflex; MACCE, major adverse cardiac and cardiovascular events.

* statistically significant (p <0.05)

In this study, we created a new score, called the J-CTO extension score, that can predict the chronic outcomes of patients post CTO-PCI. The J-CTO extension score could predict the incidence of MACCE at the 1-year follow-up, with stepwise alternations in the derivation and

**Table 3. Multivariate analysis for the incidences of MACCE and the scoring model.**

| | 95% CI | Odds ratio | p value | Score |
|---|---|---|---|---|
| **Diabetes mellitus** | 1.131–2.622 | 1.722 | 0.011* | 1 |
| **History of PAD** | 0.761–2.361 | 1.34 | 0.311 | 0 |
| **Hemodialysis** | 1.286–5.064 | 2.552 | 0.007* | 3 |
| **LVEF <35%** | 1.191–4.187 | 2.233 | 0.012* | 2 |
| **In-stent occlusion** | 1.407–3.691 | 2.279 | 0.001* | 2 |
| **Calcification** | 0.732–1.819 | 1.154 | 0.539 | 0 |
| **Tortuous lesion** | 0.946–3.091 | 1.71 | 0.076 | 0 |

CI, confidence interval; LVEF, left ventricular ejection fraction; MACCE, major adverse cardiac and cardiovascular events; PAD, peripheral artery disease.

* statistically significant (p <0.05)

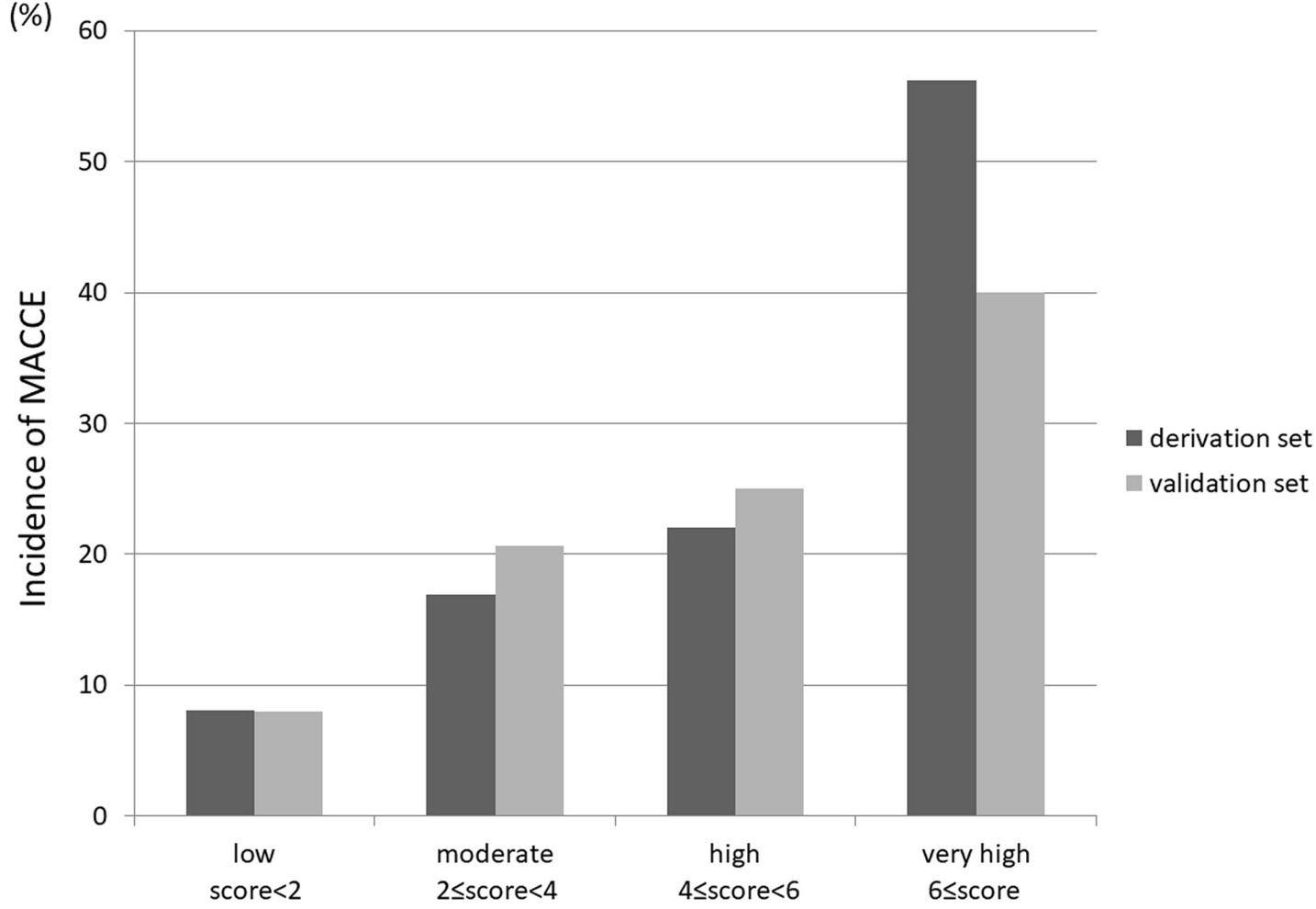

**Fig 3. Incidence of MACCE: Comparison between the derivation and validation groups.** We created a scoring model, named J-CTO extension score, according to the odds ratio of multivariate analysis of MACCE at the 1-year follow-up, as follows: hemodialysis = 3, LVEF <35% = 2, in-stent occlusion = 2, and DM = 1. This model was successful in predicting MACCE incidence at the 1-year follow-up with stepwise alterations in the derivation and validation sets. LVEF, left ventricular ejection fraction; MACCE, major advanced cardiovascular events; J-CTO, Japan-chronic total occlusion.

validation datasets. Furthermore, among other established CTO scores, only the J-CTO extension score predicted the chronic outcomes of patients post CTO-PCI. The novel scoring system was associated with the incidence of all cause death. However, the association with myocardial infarction and stroke was less clear.

The potential benefits of CTO recanalization based on observational studies and meta-analysis include: improvement in symptoms, relief of ischemia, and improvement of left ventricular function. Moreover, the long-term improvements in the clinical outcome of patients post CTO-PCI has been assessed in several reports [18–20]. Jones et al. [21] and George et al. [22] reported that successful CTO-PCI improves long-term survival compared with unsuccessful cases. Mehran et al. demonstrated that successful CTO-PCI decreased the need for coronary artery bypass graft surgery (hazard ratio: 0.21; 95% CI, 0.13–0.40; p <0.01) [23]. Our study demonstrated a sufficient predictability of the risk of developing MACCE at 1 year post CTO-PCI according to a model based solely on pre-procedural variables. These findings may lead to a better stratification of patients at risk for developing MACCE at 1 year post CTO-PCI,

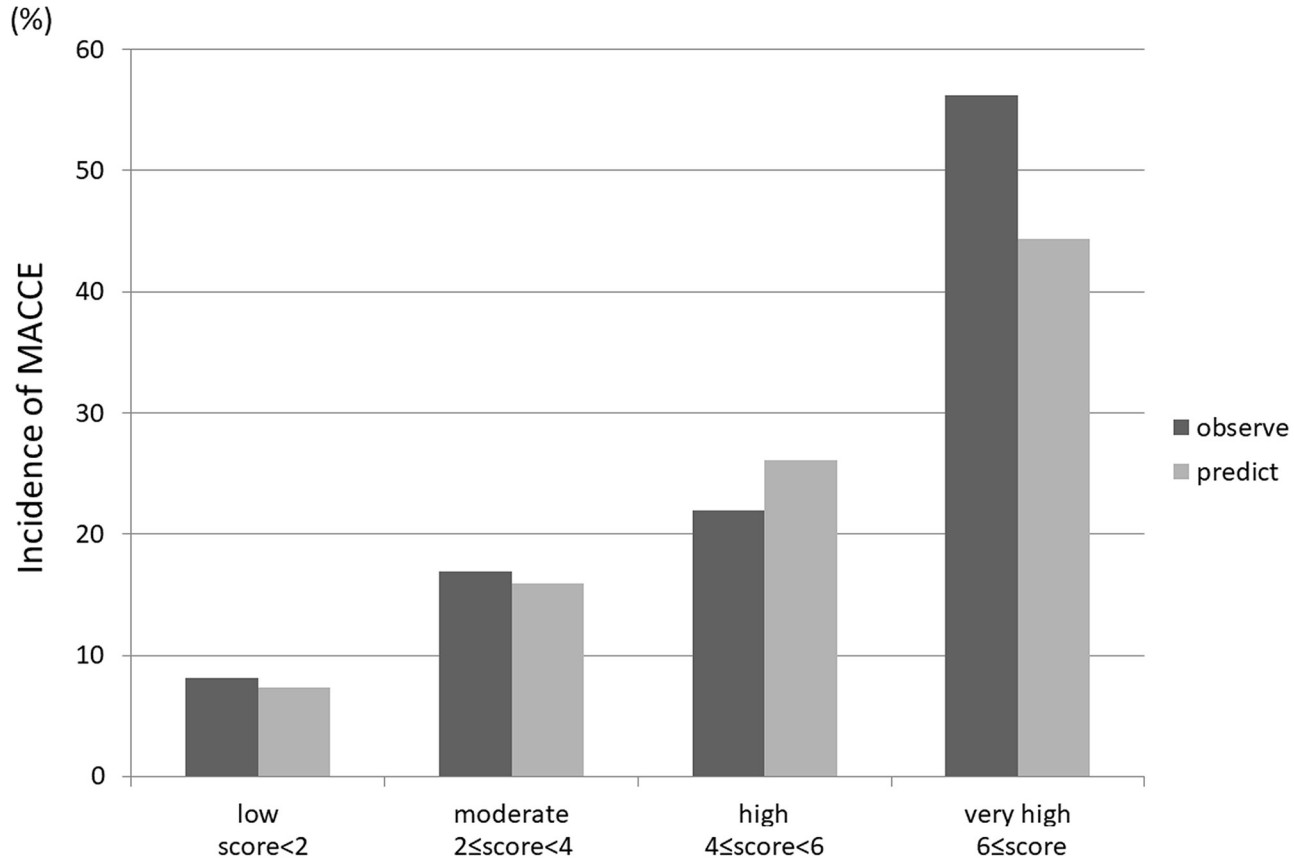

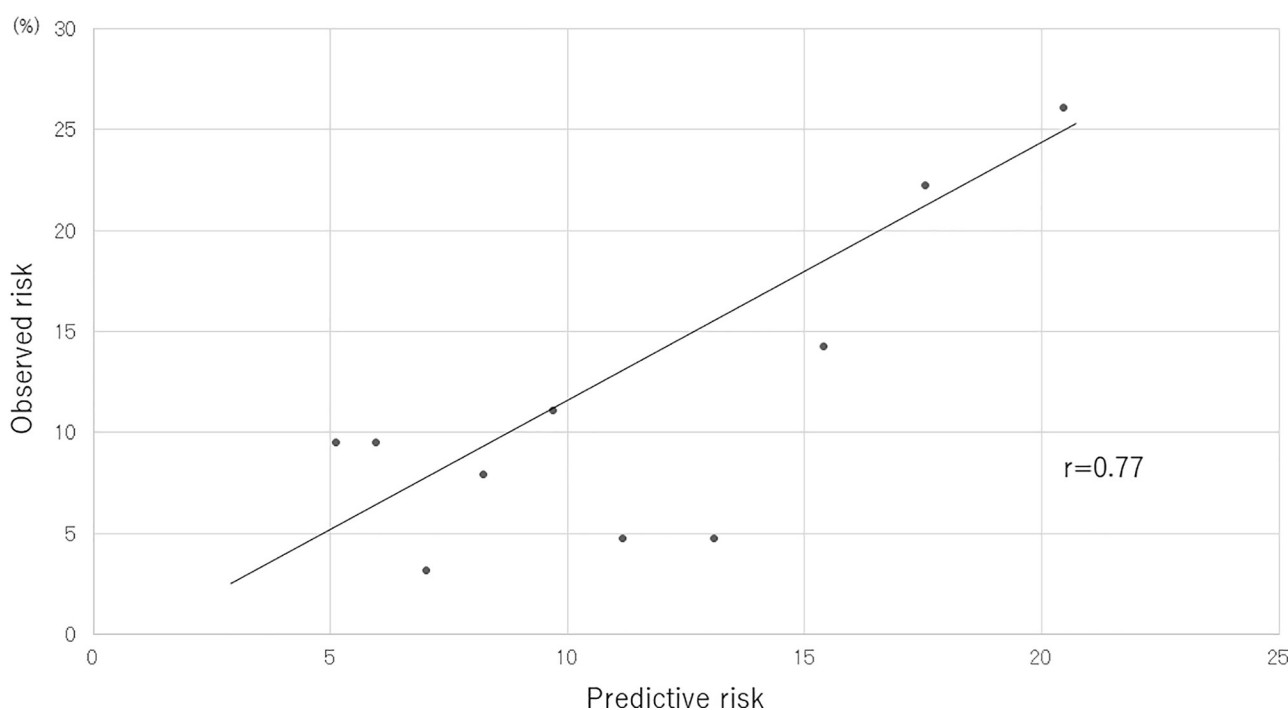

**Fig 4. Internal validation of the J-CTO extension score.** A: This figure revealed calibration of the J-CTO extension score in the validation group. The average predicted versus observed probability of MACCE with the J-CTO extension score for each quartile (categorized as low, moderate, high, and very high-risk groups) was: 8.1% vs. 7.3%, 16.9% vs. 15.9%, 22.0% vs. 26.1%, and 56.2% vs. 44.4% of the observed and predicted scores, respectively. B: The agreements between the observed and predicted risks of MACCE at the 1-year follow-up, with developed risk-scoring methods were assessed across the 10 groups divided according to the risk score in the validation cohort. There was a significant correlation between the values of the observed and predicted risk in the 10 groups (r = 0.77). MACCE, major advanced cardiovascular events; J-CTO, Japan-chronic total occlusion.

before performing other procedures. Furthermore, our study might reveal the current incident rate of developing MACCE at 1 year after performing CTO-PCI in a Japanese population.

Previous reports of successful long-term outcomes following CTO-PCI indicated that the mortality and target vessel revascularization (TVR) incidence rates at 5-year follow-up were 4.5% and 11.5%, respectively [19]. Although, the follow-up period was relatively shorter in our population, the outcomes might be similar to that of the previous report, based on our Kaplan-Meier analysis (the death and TVR incidence rates were 2.3% and 8.4% at 1 year post PCI, respectively). Previous investigations of the long-term prognosis following CTO-PCI have focused on comparisons of successful vs. unsuccessful cases and were not designed to assess the long-term prognosis using pre-procedural variables. With our novel strategy, approximately 80% of easy CTO-PCI cases could be passed with a single guidewire and, thus, have a shorter procedural time [24]. Conversely, difficult cases of CTO remain a challenge, and their procedural success rate is lower than that of easy cases [25,26], even when the procedures were performed by expert operators. Thus, owing to the difficulty in conducting the interventional procedures, the initial success rate of CTO-PCI has been inferior to that of CABG [4]. To select between the CTO-PCI and CABG procedures, we should consider the initial success rates and chronic outcomes, including revascularization. The performance of the PROGRESS CTO score for the prediction of long-term outcomes was tested recently, and the score was associated with the MACCE risk, albeit vigorous statistical adjustment was not performed [13]. To clarify the risk estimation of MACCE and refining the indication for CTO-PCI, a universal risk stratification was needed for assessing CTO-PCI as a suitable option for the treatment of ischemic heart disease. Although CTO-PCI has a potential of providing several benefits to patients with coronary artery disease, some of them might not receive such benefits, although the initial procedural success was obtained. Thus, it is important to identify a group of patients in whom CTO-PCI could contribute to their long-term outcome and use this information for the decision making of their therapeutic strategy, despite the difficulty in estimating the long-term outcome before an uncertain intervention like CTO-PCI. In our population, the previous CTO-PCI scoring systems could not predict the chronic outcomes of CTO-PCI (Fig 5). A specialized scoring model is required to predict the chronic outcome of CTO-PCI. This novel scoring system might help in choosing the appropriate revascularization method for patients with CTO lesions, just as the previous scores could help in estimating the initial success of CTO-PCI.

The main strength of our study was the development of a simple scoring model based on common clinical information. The association of the MACCE incidence and the cumulative effect of the variable listed in the score was linear and these variables were independently associated with the risk of MACCE development. The unique demographic characteristics of the patients with incidences of MACCE at 1 year follow-up were also revealed in this study, including the independent predictors of incidences of MACCE at 1 year follow-up (three clinical variables [e.g., hemodialysis, low LVEF, and diabetes mellitus] and one lesion-related variable [in-stent occlusion]). In this population, 1-year survival, non-fatal myocardial infarction, and non-fatal stroke were not different between the in-stent and not in-stent occlusion groups; however, target vessel revascularization was significantly higher in the in-stent occlusion

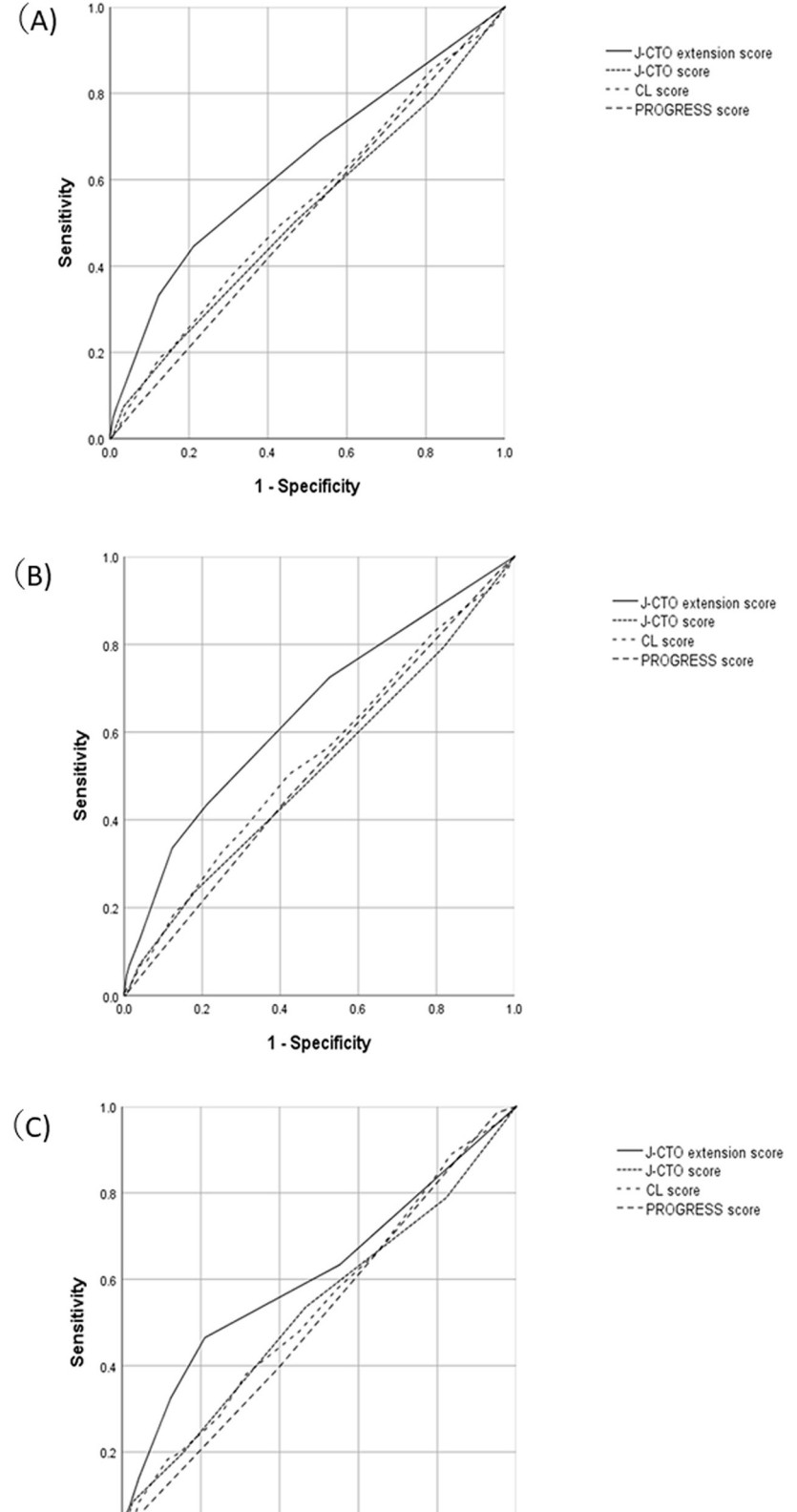

**Fig 5. Comparison between the J-CTO extension score and other scoring models of CTO-PCI.** A: ROC curve of the overall population. The AUC of the J-CTO, CL, and PROGRESS scores was 0.518 (95% CI, 0.473–0.563, p = 0.406), 0.540 (95% CI, 0.496–0.583, p = 0.066), and 0.514 (95% CI, 0.472–0.556, p = 0.509), respectively. Conversely, the J-CTO extension score was only associated with the incidence of MACCE at the 1-year follow-up; the AUC was 0.634 (95% CI, 0.590–0.678, p <0.0001). B: ROC curve of the derivation group. The AUC of the J-CTO, CL, and PROGRESS scores was 0.514 (95% CI, 0.459–0.569, p = 0.600), 0.543 (95% CI, 0.489–0.597, p = 0.105), and 0.515 (95% CI, 0.463–0.567, p = 0.585), respectively. Conversely, the J-CTO extension score was only associated with the incidence of MACCE at the 1-year follow-up; the AUC was 0.650 (95% CI, 0.598–0.703, p <0.0001). 4C: ROC curve of the validation group. The AUC the J-CTO, CL, and PROGRESS scores was 0.528 (95% CI, 0.452–0.604, p = 0.44), 0.535 (95% CI, 0.463–0.608 p = 0.33), and 0.512 (95% CI, 0.442–0.581, p = 0.751), respectively. Conversely, the J-CTO extension score was only associated with the incidence of MACCE at the 1-year follow-up; the AUC was 0.610 (95% CI, 0.532–0.688, p <0.003). CI, confidence interval; AUC, area under the curve, ROC, receiver operating characteristic; J-CTO, Japan-chronic total occlusion score; MACCE, major advanced cardiovascular events.

group (p <0.0001). Although the follow-up rate was low, this study benefited from the high initial success rate of CTO-PCI amongst experienced Japanese operators (88.3% and 87.5% procedural and patient success rates, respectively).

However, our study had several limitations. First, core laboratory assessments were not performed. Therefore, all data, including the angiographic data, were obtained from the operators through self-report. Second, the occluded period could not be identified for more than 70% of the population, which is an issue often encountered in CTO research. Third, the low follow-up rate for patients is a significant concern. Sometimes, patients with CTO lesions were transferred from a small institution to a high-volume center; therefore, the patients would often be followed up by another institution after the procedure, which might be the cause of the low follow-up rate. Further analysis might be required in populations with high follow-up rates. Fourth, the follow-up period was relatively short. Finally, an antegrade dissection and the reentry system was not used, which might affect the outcome of CTO-PCI.

## Conclusions

In this study we presented the clinical and angiographic parameters that predict the outcome in patients who had undergone CTO-PCI. Their data were obtained from a Japanese multicenter registry and were configured in the creation of a novel scoring model using a logistic regression approach. This model allows the identification of four subgroup scores corresponding to very-high, high, moderate, and low incidences of MACCE at the 1-year follow-up following CTO-PCI. The increased scores were correlated with a high probability of MACCE incidence at the 1-year follow-up, ranging from 5% to more than 35%. The J-CTO extension score was the only scoring system to predict MACCE outcomes following CTO-PCI. However, the applicability of the J-CTO extension score should be validated in other centers.

## Supporting information

**S1 Study protocol.**
(DOCX)

## Acknowledgments

We are grateful to the members of the cardiac catheterization laboratories of the participating centers and the clinical research coordinators.

## Author Contributions

**Conceptualization:** Soichiro Ebisawa, Maoto Habara.

**Data curation:** Soichiro Ebisawa, Toshiya Muramatsu, Yoshifumi Kashima, Atsunori Oka-mura, Masahisa Yamane, Masami Sakurada, Shunsuke Matsuno, Mikihiro Kijima, Maoto Habara.

**Formal analysis:** Soichiro Ebisawa.

**Investigation:** Soichiro Ebisawa.

**Supervision:** Shun Kohsaka.

**Writing – original draft:** Soichiro Ebisawa.

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
