## [Decision Letter · Decision Letter 0]

1 May 2020

PONE-D-20-00743

Derivation and validation of the J-CTO extension score for pre-procedural prediction of major adverse cardiac and cerebrovascular events in patients with chronic total occlusions

PLOS ONE

Dear Mr. Ebisawa,

Thank you for submitting your manuscript to PLOS ONE. After careful consideration, we feel that it has merit but does not fully meet PLOS ONE’s publication criteria as it currently stands. Therefore, we invite you to submit a revised version of the manuscript that addresses the points raised during the review process.

The reviewer pointed out the problems, so please response them soon.

We would appreciate receiving your revised manuscript by Jun 15 2020 11:59PM. To enhance the reproducibility of your results, we recommend that if applicable you deposit your laboratory protocols in protocols.io, where a protocol can be assigned its own identifier (DOI) such that it can be cited independently in the future. For instructions see: http://journals.plos.org/plosone/s/submission-guidelines#loc-laboratory-protocols

We look forward to receiving your revised manuscript.

Kind regards,

Yoshiaki Taniyama, MD, PhD

Academic Editor

PLOS ONE

Journal Requirements:

2.Thank you for including your ethics statement:

"This study was approved by the review board of each institution, and written informed consent was obtained from all patients."

Please clarify the sources of funding (financial or material support) for your study. List the grants or organizations that supported your study, including funding received from your institution.If any authors received a salary from any of your funders, please state which authors and which funders.If you did not receive any funding for this study, please state: “The authors received no specific funding for this work.”

"I have read the journal's policy and the authors of this manuscript have the following competing interests: Soichiro Ebisawa; Affiliation with Endowed Department…Medtronic, Abbott Vascular Japan, Boston Scientifics Japan, TERUMO, NIPRO and Cordis."

We note that one or more of the authors are employed by commercial companies: Medtronic, Abbott Vascular Japan, Boston Scientifics Japan, TERUMO, NIPRO.

6. Please amend your list of authors on the manuscript to ensure that each author is linked to an affiliation. Authors’ affiliations should reflect the institution where the work was done (if authors moved subsequently, you can also list the new affiliation stating “current affiliation:….” as necessary).

7. Please include captions for your Supporting Information files at the end of your manuscript, and update any in-text citations to match accordingly. Please see our Supporting Information guidelines for more information: http://journals.plos.org/plosone/s/supporting-information

Reviewers' comments:

Reviewer's Responses to Questions

**Comments to the Author**

1. Is the manuscript technically sound, and do the data support the conclusions?

Reviewer #1: Yes

2. Has the statistical analysis been performed appropriately and rigorously? 

Reviewer #1: Yes

3. Have the authors made all data underlying the findings in their manuscript fully available?

Reviewer #1: No

4. Is the manuscript presented in an intelligible fashion and written in standard English?

Reviewer #1: Yes

5. Review Comments to the Author

Reviewer #1: I have read with great interest the manuscript entitled “Derivation and validation of the J-CTO extension score for pre-procedural prediction of major adverse cardiac and cerebrovascular events in patients with chronic total occlusions”. The paper makes a significant contribution to the understanding of potential risk factors affecting the long-term prognosis of patients undergoing PCI for CTO. The article is well written, comprises a large number of patients treated at experienced Japanese centres, and a robust methodology provides a new perspective on the problem. Yet, a couple of issues should be elaborated or explained.

Comment 1

The inclusion/exclusion criteria seem unclear since the authors claim that they excluded from the baseline cohort (4909 patients) 770 subjects due to the lack of one-year follow-up data. From the remaining 4139 patients, they excluded another 2230, again because of the lack of one-year follow-up;

Comment 2

Very few data on PCI techniques were presented in the document. The authors should submit at least the percentage of patients revascularized with antegrade and retrograde technique. Also, it is desirable to mention if the PCI technique had any influence on the clinical outcomes;

Comment 3

One of the most surprising observations is the fact that the initial success of the procedure was not associated with the incidence of one-year MACCE. The authors just mention this very casually; I believe they should present more data on this, at least respective percentages in Table 2;

Comment 4

Another remarkable finding was that in-stent occlusion lesions were associated with the incidence of MACCE; I believe the authors should elaborate on this observation in the Discussion section;

Comment 5

Finally, recently The EuroCTO (CASTLE) Score has been published. Would it be possible to analyze its predictive value, just as the authors did for two other scores (PROGRES and CL score?

6. PLOS authors have the option to publish the peer review history of their article (what does this mean?). If published, this will include your full peer review and any attached files.

Reviewer #1: No

---

## [Author Response · Author response to Decision Letter 0]

11 May 2020

Dear Dr. Yoshiaki Taniyama:

Ref. No.: PONE-D-20-00743

Title: Derivation and Validation of the PJ-CTO Score for Pre-Procedural Prediction of Major Coronary and Cerebrovascular Events in Patients with Chronic Total Occlusions.

I, along with my coauthors, would like to re-submit the attached manuscript as an Original Article.

The manuscript has been carefully rechecked and appropriate changes have been made in accordance with the reviewers’ suggestions. The responses to their comments have been prepared and attached herewith. 

We thank you and the reviewers for your thoughtful suggestions and insights, which have enriched the manuscript and produced a more balanced and better account of the research. We hope that the revised manuscript is now suitable for publication in your esteemed PLOS ONE:

We look forward to hearing from you at your earliest convenience.

Sincerely,

Soichiro Ebisawa, MD, PhD 

Department of Cardiovascular Medicine, Shinshu University School of Medicine 

3-1-1 Asahi, Matsumoto-shi 390-8621, Japan

Tel: +81-263-37-3486/Fax: +81-263-37-3489

Email: ebisawa@shinshu-u.ac.jp

Comments from the editors and reviewers:

Reviewer #1: I have read with great interest the manuscript entitled “Derivation and validation of the J-CTO extension score for pre-procedural prediction of major adverse cardiac and cerebrovascular events in patients with chronic total occlusions”. The paper makes a significant contribution to the understanding of potential risk factors affecting the long-term prognosis of patients undergoing PCI for CTO. The article is well written, comprises a large number of patients treated at experienced Japanese centres, and a robust methodology provides a new perspective on the problem. Yet, a couple of issues should be elaborated or explained.

Response

We would like to express our appreciation to the Reviewer for his/her insightful comments, which have helped us to improve the report significantly.

Comment 1

The inclusion/exclusion criteria seem unclear since the authors claim that they excluded from the baseline cohort (4909 patients) 770 subjects due to the lack of one-year follow-up data. From the remaining 4139 patients, they excluded another 2230, again because of the lack of one-year follow-up;

Response

Indeed, I thought it was unclear and difficult to understand the meaning of sentence. It means 770 patients who underwent procedures in institutions that did not join collect follow-up one-year data, and I excluded such institution’s data at first. However, 2230 patients still had no data of one-year follow up and should be excluded.

According to the reviewer’s comment, we change the sentences as “did not join to collect one-year follow-up data” in line 16 page 6.

Comment 2

Very few data on PCI techniques were presented in the document. The authors should submit at least the percentage of patients revascularized with antegrade and retrograde technique. Also, it is desirable to mention if the PCI technique had any influence on the clinical outcomes;

Response

The reviewer raise an important point.

According to the reviewer’s comment, we add the sentences “Regarding with technical issue, retrograde approach was performed in 606 cases (31.7%), however this procedure was not associated with both initial success (p=0.569) and incidence of one-year MACCE (p=0.368)” in line 11 page 10.

Comment 3

One of the most surprising observations is the fact that the initial success of the procedure was not associated with the incidence of one-year MACCE. The authors just mention this very casually; I believe they should present more data on this, at least respective percentages in Table 2;

Response

CTO-PCI has been improved latest 10 years, however initial success might not necessarily reflect the long-term outcome. We think this comment is very important in this field, and we should establish another registry to assess this point.

According to the reviewer’s comment, we add the correlation between initial procedural success and incidence of one-year MACCE in table 2.

Comment 4

Another remarkable finding was that in-stent occlusion lesions were associated with the incidence of MACCE; I believe the authors should elaborate on this observation in the Discussion section;

Response

This comment is also important. Long term outcome of PCI after in stent occlusion case was not favorable. We should assess the indication of revascularization of in stent occlusion lesion.

According to reviewer’s comment, we add the sentence of “In this population, one-year survival, non-fatal myocardial infarction and non-fatal stroke were not different between in-stent occlusion and not in-stent occlusion group, however target vessel revascularization was significantly higher in in-stent occlusion group (p<0.0001)” in line 8 page 23.

Comment 5

Finally, recently The EuroCTO (CASTLE) Score has been published. Would it be possible to analyze its predictive value, just as the authors did for two other scores (PROGRES and CL score?

Response

When, first time, we submit this manuscript, CASTLE score was not established. We also think this score is well-established score in big population data. According to the comment we compare between J-CTO extension score and CASTLE by ROC analysis, CASTLE score was not associated with one-year MACCE (AUC=0.557), while J-CTO extension score was associated with one-year MACCE (AUC=0.647).

---

## [Decision Letter · Decision Letter 1]

4 Jun 2020

PONE-D-20-00743R1

Derivation and validation of the J-CTO extension score for pre-procedural prediction of major adverse cardiac and cerebrovascular events in patients with chronic total occlusions

PLOS ONE

Dear Dr. Ebisawa,

Thank you for submitting your manuscript to PLOS ONE. After careful consideration, we feel that it has merit but does not fully meet PLOS ONE’s publication criteria as it currently stands. Therefore, we invite you to submit a revised version of the manuscript that addresses the points raised during the review process.

We did statistical review, and got some problems. Please response to them.

We look forward to receiving your revised manuscript.

Kind regards,

Yoshiaki Taniyama, MD, PhD

Academic Editor

PLOS ONE

Reviewers' comments:

Reviewer's Responses to Questions

**Comments to the Author**

1. If the authors have adequately addressed your comments raised in a previous round of review and you feel that this manuscript is now acceptable for publication, you may indicate that here to bypass the “Comments to the Author” section, enter your conflict of interest statement in the “Confidential to Editor” section, and submit your "Accept" recommendation.

Reviewer #1: All comments have been addressed

Reviewer #2: (No Response)

2. Is the manuscript technically sound, and do the data support the conclusions?

Reviewer #1: Yes

Reviewer #2: Partly

3. Has the statistical analysis been performed appropriately and rigorously? 

Reviewer #1: Yes

Reviewer #2: No

4. Have the authors made all data underlying the findings in their manuscript fully available?

Reviewer #1: Yes

Reviewer #2: No

5. Is the manuscript presented in an intelligible fashion and written in standard English?

Reviewer #1: Yes

Reviewer #2: Yes

6. Review Comments to the Author

Reviewer #1: The authors responded to my comments in a satisfactory manner, and I believe that the paper is suitable for publication in its current form.

Reviewer #2: PONE-D-20-00743R1: statistical review

SUMMARY. This paper investigates factors that increase the risk of developing major adverse events (MACCE) among patients with chronic total occlusion who undergo percutaneous coronary intervention (PCI). The statistical analysis is made in two steps. First a battery of logistic regression models is estimated to identify the significant predictors of MACCE. Then the results of the logistic regression are used to define a risk score. I have little to say about the first step: logistic regression is the correct methodology to identify risk factors and it is appropriately developed in the paper. I have however some concerns about the second step (see major issues below). I also list below some specific points that should be addressed.

MAJOR ISSUES

1) The multivariate logistic regression estimates provided in Table 3 provide an optimal way to estimate the probability of a MACCE, through the formula

Prob(MACCE01)= exp(beta_0+beta_1x_1+ ... \\beta_Kx_k)/(1+exp(beta_0+beta_1x_1+ ... \\beta_Kx_k))

where the betas are log-ORs. The proposed score makes sense only if it able to beat the predictive power of the multivariate logistic regression. The only way to prove this is to fit a logistic regression model

logit(Prob(MACCE=1))=beta_0+beta_1 score and compare this model to the multivariate regression of Table 3.

2) Line 8, page 16 "We created a scoring model (J-CTO Extension score) according to the odds ratio". I was not able to understand the rule that has been used to transform the odds ratios into the scores. Please clarify.

3) If I understood correctly (not clear from text), the score is defined by summing the scores in the last column of Table 3. This makes sense if the cumulative effects of hemodialysis, LVEF<35%, in-stent occlusion and DM=1 is additive. However, the multivariate logistic regression of Table 3 clearly tells another story, that the cumulative effect of these factors is nonlinear.

SPECIFIC ISSUES

1) Please clarify that MACCE means "major adverse cardiac and cerebrovascular event" in the abstract.

2) The phrase "the area under the curve" appears often in both the abstract and the main text: it should be replaced by "the are under the ROC curve".

3) page 9, line 15: why "paired" t-test? I guess that the analysis is not based on paired observations. Please check.

4) From Figures 2 and 3A, it seems that the proposed score is treated as a categorical covariate. Why? Cutting a continuous variable in classes is a unnecessary waste of information and it is not generally recommended.

5) I was not able to understand what figure 3B represents. How were the predicted and the observed risks computed? Please clarify.

6) Although the authors declare that the data are fully available without restriction, there is no data attached. The data should be attached to the paper as supplementary file to allow results reproducibility.

7. PLOS authors have the option to publish the peer review history of their article (what does this mean?). If published, this will include your full peer review and any attached files.

Reviewer #1: No

Reviewer #2: No

---

## [Author Response · Author response to Decision Letter 1]

4 Aug 2020

July 15th, 2020

Dr. Yoshiaki Taniyama

Academic Editor 

PLOS ONE

Dear Dr. Yoshiaki Taniyama:

Ref. No.: PONE-D-20-00743

Title: Derivation and Validation of the PJ-CTO Score for Pre-Procedural Prediction of Major Coronary and Cerebrovascular Events in Patients with Chronic Total Occlusions.

I, along with my coauthors, would like to re-submit the attached manuscript as an Original Article in PLOS ONE. The manuscript has been carefully rechecked and appropriate changes have been made in accordance with the reviewers’ suggestions. In particular, we have addressed all the reviewer's comments in a point-by-point manner and revisions are indicated in red font in the revised paper. We would like to thank you and the reviewers for your thoughtful suggestions and insights, which have enriched the manuscript and produced a more balanced and better account of the research. 

We hope that the revised manuscript is viewed favorably. We look forward to hearing from you.

Sincerely,

Soichiro Ebisawa, MD, PhD 

Department of Cardiovascular Medicine, Shinshu University School of Medicine 

3-1-1 Asahi, Matsumoto-shi 390-8621, Japan

Tel: +81-263-37-3486/Fax: +81-263-37-3489

Email: ebisawa@shinshu-u.ac.jp

 

Reviewer #2: PONE-D-20-00743R1: statistical review

SUMMARY. This paper investigates factors that increase the risk of developing major adverse events (MACCE) among patients with chronic total occlusion who undergo percutaneous coronary intervention (PCI). The statistical analysis is made in two steps. First a battery of logistic regression models is estimated to identify the significant predictors of MACCE. Then the results of the logistic regression are used to define a risk score. I have little to say about the first step: logistic regression is the correct methodology to identify risk factors and it is appropriately developed in the paper. I have however some concerns about the second step (see major issues below). I also list below some specific points that should be addressed.

MAJOR ISSUES

1) The multivariate logistic regression estimates provided in Table 3 provide an optimal way to estimate the probability of a MACCE, through the formula

Prob(MACCE01)= exp(beta_0+beta_1x_1+ ... \\beta_Kx_k)/(1+exp(beta_0+beta_1x_1+ ... \\beta_Kx_k))

where the betas are log-ORs. The proposed score makes sense only if it able to beat the predictive power of the multivariate logistic regression. The only way to prove this is to fit a logistic regression model

logit(Prob(MACCE=1))=beta_0+beta_1 score and compare this model to the multivariate regression of Table 3.

Response

We appreciate the reviewer’s thoughtful suggestion. Following his/her comment, we constructed the multivariate logistic regression model and compared its performance with our scoring system. When the C-stats were calculated for each model, no statistically significant difference between the two models was observed (0.665 vs. 0.658 for the scoring system and multivariate logistic regression model, respectively). We believe that our scoring system has an advantage over the traditional logistic regression model, given, its simplicity and relevance of chosen variables in clinical practice. 

Accordingly, we have added the following sentence to the revised manuscript (page 16, line 15) 

“To assess the predictive power of this model, the proposed performance scoring system was compared with that of the traditional multivariate regression model (covariates selected from variables in Table 3) and only a small difference in C-statistics between them was observed (0.665 vs. 0.658).”

2) Line 8, page 16 "We created a scoring model (J-CTO Extension score) according to the odds ratio". I was not able to understand the rule that has been used to transform the odds ratios into the scores. Please clarify.

Response

We would like to thank the reviewer for the comment. To construct our scoring model, identified variables (from traditional logistic regression analyses) were assigned a weighted integer based on the calculated odds ratio. This method has been validated in previous reports. For example, R. Mehran, et al. demonstrated that this integer scoring system was useful in prediction of contrast induced nephropathy (J Am Coll Cardiol 2004;44:1393-1389). More recently, the National Cardiovascular Data Registry (from the American College of Cardiology) developed risk estimation models using the same method (J Am Coll Cardiol 2010;55:1923-32 and J Am Coll Cardiol 2013;6:790-799). These methods are simple and perhaps user-friendly. We have changed the aforementioned sentence, as follows:

“We created a scoring model (J-CTO extension score) by assigning a weighted integer based on the calculated OR”

3) If I understood correctly (not clear from text), the score is defined by summing the scores in the last column of Table 3. This makes sense if the cumulative effects of hemodialysis, LVEF<35%, in-stent occlusion and DM=1 is additive. However, the multivariate logistic regression of Table 3 clearly tells another story, that the cumulative effect of these factors is nonlinear.

Response

We would like to thank the reviewer for the valuable comments. The reviewer correctly stated that the score is defined by summing the scores listed in Table 3. To address the reviewer’s concern, we have directly calculated the incidence of MACCE and number of the variables group (variable number = 0, 1, 2 and 3 or 4). 

Figure A

As demonstrated in Figure A, the association of MACCE incidence and cumulative effect of the variable listed in the score was indeed linear. We have also looked at the distribution of the aforementioned variables among our patients (Figure B). The majority of the patients with the above variables (hemodialysis [HD], in-stent occlusion [ISO], and low LVEF) did not overlap in patients with or without diabetes (DM).

Figure B

Please also note that HD, in stent occlusion, low LVEF, and DM were independently associated with risk of MACCE within our original logistic regression analysis. We have also compared the performance of this inter scoring system with a traditional logistic regression model (in response to Comment #1), and the c-stats were almost equivocal.

Please note that we have added the following sentence in our revised manuscript (line 14 page 23

“The association of the MACCE incidence and the cumulative effect of the variable listed in the score was linear and these variables were independently associated with the risk of MACCE development.” 

SPECIFIC ISSUES

1) Please clarify that MACCE means "major adverse cardiac and cerebrovascular event" in the abstract.

Response

According to the reviewer’s point, we have added the sentence “major adverse cardiac and cerebrovascular event” in the Abstract section.

2) The phrase "the area under the curve" appears often in both the abstract and the main text: it should be replaced by "the are under the ROC curve".

Response

Following the reviewer’s suggestion, we have replaced the phrase “the area under the curve” with “the area under the ROC” in the manuscript.

3) page 9, line 15: why "paired" t-test? I guess that the analysis is not based on paired observations. Please check.

Response

We agree with the reviewer. According to the reviewer’s suggestion, we have revised this point from paired to unpaired.

4) From Figures 2 and 3A, it seems that the proposed score is treated as a categorical covariate. Why? Cutting a continuous variable in classes is a unnecessary waste of information and it is not generally recommended.

Response

The reviewer raised an important point which we agree with. 

One of the purposes of this study was to assess the risk stratification of CTO-PCI. To assess the risk of each case simpler, we evaluated the four-risk groups.

5) I was not able to understand what figure 3B represents. How were the predicted and the observed risks computed? Please clarify.

Response

We appreciate the reviewer’s important suggestion. 

For calibration of this scoring model performance, we represented Figure 3B.

We have divided the validation cohort into 10 groups according to the risk score and calculated the predicted risks by logistic regression analysis. Then, we compared the observed and the predicted risks in each group.

We have revised the correlation between the observed and predicted risks and there was significant correlation between of them (r = 0.77).

According to this result, we have replaced the Figure 4B in the revised manuscript (below).

Figure 4B

6) Although the authors declare that the data are fully available without restriction, there is no data attached. The data should be attached to the paper as supplementary file to allow results reproducibility.

Response

The data and materials used to conduct this research are available to the researchers to reproduce the results or replicate the procedure on request. The procedure does need to follow the Act on the Protection of Personal Information Law (as of May 2017) and the Ethical Guidelines for Medical and Health Research Involving Human Subjects (as of March 2015) in Japan.

---

## [Decision Letter · Decision Letter 2]

21 Aug 2020

Derivation and validation of the J-CTO extension score for pre-procedural prediction of major adverse cardiac and cerebrovascular events in patients with chronic total occlusions

PONE-D-20-00743R2

Dear Dr. Ebisawa,

We’re pleased to inform you that your manuscript has been judged scientifically suitable for publication and will be formally accepted for publication once it meets all outstanding technical requirements.

Kind regards,

Yoshiaki Taniyama, MD, PhD

Academic Editor

PLOS ONE

Additional Editor Comments (optional):

Reviewers' comments:

Reviewer's Responses to Questions

**Comments to the Author**

1. If the authors have adequately addressed your comments raised in a previous round of review and you feel that this manuscript is now acceptable for publication, you may indicate that here to bypass the “Comments to the Author” section, enter your conflict of interest statement in the “Confidential to Editor” section, and submit your "Accept" recommendation.

Reviewer #2: All comments have been addressed

2. Is the manuscript technically sound, and do the data support the conclusions?

Reviewer #2: (No Response)

3. Has the statistical analysis been performed appropriately and rigorously? 

Reviewer #2: (No Response)

4. Have the authors made all data underlying the findings in their manuscript fully available?

Reviewer #2: (No Response)

5. Is the manuscript presented in an intelligible fashion and written in standard English?

Reviewer #2: (No Response)

6. Review Comments to the Author

Reviewer #2: (No Response)

7. PLOS authors have the option to publish the peer review history of their article (what does this mean?). If published, this will include your full peer review and any attached files.

Reviewer #2: No

---

## [Editor Report · Acceptance letter]

25 Aug 2020

PONE-D-20-00743R2 

Derivation and validation of the J-CTO extension score for pre-procedural prediction of major adverse cardiac and cerebrovascular events in patients with chronic total occlusions 

Dear Dr. Ebisawa:

I'm pleased to inform you that your manuscript has been deemed suitable for publication in PLOS ONE. Congratulations! Your manuscript is now with our production department. 

Kind regards, 

on behalf of

Dr Yoshiaki Taniyama 

Academic Editor

PLOS ONE